# Assessment of the Body Composition and Bone Calcification of Students of Police Schools and Police Training Centers in Poland—A Cross-Sectional Study

**DOI:** 10.3390/ijerph19127161

**Published:** 2022-06-10

**Authors:** Tomasz Lepionka, Anna Anyżewska, Ewelina Maculewicz, Krzysztof Klos, Roman Lakomy, Ewa Szarska, Andrzej Tomczak, Agata Gaździńska, Katarzyna Skuza, Jerzy Bertrandt

**Affiliations:** 1Military Institute of Hygiene and Epidemiology, 4 Kozielska, 01-163 Warsaw, Poland; kklos@wim.mil.pl (K.K.); roman.lakomy@interia.pl (R.L.); eszarska@gmail.com (E.S.); katarzyna.skuza@wihe.pl (K.S.); 2University of Economics and Human Sciences in Warsaw, Okopowa 59, 01-043 Warsaw, Poland; a.anyzewska@vizja.pl; 3Faculty of Physical Education, Józef Pilsudski University of Physical Education in Warsaw, 34 Marymoncka, 00-968 Warsaw, Poland; ewelina.maculewicz@awf.edu.pl; 4Independent Researcher, 02-348 Warsaw, Poland; biuro.at@onet.pl; 5Laboratory of Dietetics and Obesity Treatment, Department of Psychophysiological Measurements and Human Factor Research, Military Institute of Aviation Medicine, 54/56 Krasinskiego, 01-755 Warsaw, Poland; afrotena@gmail.com; 6Faculty of Economic Sciences, John Paul II University of Applied Sciences in Biala Podlaska, Sidorska 95/97, 21-500 Biala Podlaska, Poland; jwbertrandt@gmail.com

**Keywords:** police schools, body composition, obesity, osteoporosis, osteopenia, DEXA

## Abstract

The 21st century is considered the age of malnutrition resulting in the unprecedented frequency of civilization diseases. Among these disorders, obesity is particularly distinguished and considered an epidemic-scale disease. For this reason, conducting studies on obesity and counteracting this phenomenon is essential. Research from recent years indicates a problem of excessive body weight among officers of uniformed services, who should be characterized by good health and fitness level due to the specificity of the work. As the problem of obesity affects every fourth Pole, research in uniformed services seems to be essential from health and national security perspectives. The presented study aimed to determine the elements of nutritional status in 289 students of Polish police schools and police training centers. Body composition was determined by bioelectrical impedance analysis, and bone calcification assessment was conducted by the DXA densitometric method. Based on BMI and body fat content, body weight disorders were found in 31.8% of all examined students. Densitometric test results showed changes in bone calcification of varying severity in 26.6% of the total number of respondents. The presence of obesity in students of police schools and training centers proves that the present nutrition model is energetically unbalanced, while the demonstrated disorders of bone calcification indicate an improper condition of mineral nutrition.

## 1. Introduction

A multitude and variability of factors influencing food consumption necessitate their monitoring as significant for human nutritional status. Nutritional status is a component of a health condition resulting from the habitual consumption of food, the absorption and use of its nutrients, and possible pathological factors that affect these processes [1]. The purpose of nutritional status assessment is to identify people who exhibit malnourished nutritional disorders or are overweight and obese. Overweight and obesity are common health problems in 21st century societies. The prevalence of overweight and obesity in developed countries, including Poland, has steadily increased and has been one of the most common lifestyle-related health problems [2,3]. Globally, it is estimated that 44% of adults and 20% of children over five are overweight or obese. Moreover, since 1975, the number of overweight and obese people has tripled and now accounts for 4 million deaths worldwide each year, of which nearly 2/3 are due to cardiovascular disease [4]. Currently, obesity is perceived as aesthetically problematic and, primarily, as a severe medical issue and pathophysiological disorder caused by a superabundant accumulation of adipose tissue [5]. Excessive fat accumulation leads to negative consequences for the quality of life and wellbeing, life expectancy and treatment costs [6,7], and an increased incidence of chronic non-communicable diseases [8].

One of the methods of body weight monitoring is the use of body mass index—BMI—which is the quotient of body weight (measured in kilograms) and the square of the height (in meters). It is widely used in public health and clinical nutrition to quickly evaluate nutritional wellbeing, as excessive BMI is associated with increased risks of mortality, cardiovascular disease, and some cancers. According to the standards adopted by the World Health Organization (WHO), overweight is diagnosed when the BMI value is 25 or more, and obesity is diagnosed when the value of this index is 30 or more [9]. Although BMI has a significant limitation in the discrimination between adipose tissue and muscle mass, it is included in several widely used nutritional screening tools [10].

The complexity of external and innate factors influencing nutritional status is evident. The direct cause of overweight and obesity associated with the accumulation of adipose tissue is an imbalance in energy management, caused by an excessive intake of calories from food and drinks. It is favored by other factors, i.e., an inactive lifestyle, the popularity of food products with high sugar content, new technologies reducing physical effort, environmental and genetic factors, and family habits. However, it can be concluded that nutritional status results from the applied nutrition model and physical load. The nutritional status assessment indicates whether the physiological needs of the subject in relation to the nutrient requirements are met. Numerous observations of the relationship between anthropometric measurements and nutrition have shown that such values as height and weight or body fat content are reliable indicators of nutritional status. In turn, the proper nutritional status indicates that the demand for other nutrients is sufficiently covered. Moreover, the assessment of the nutritional status allows the determination of the current state of health and is a factor in forecasting the possible development of several metabolic diseases that are directly or indirectly related to nutrition.

Research conducted in the last decade shows that obesity and the ignorance of soldiers and police officers about a healthy lifestyle are severe problems for uniformed services. It should be emphasized that these are professional groups on which the state’s security depends. Our previous study conducted on 479 police officers showed the presence of excessive body weight resulting from abdominal obesity in over 50% of male police officers [11]. A similar study conducted on 7000 Polish soldiers showed that cardiovascular diseases among military personnel pose a more significant threat in this group than among civilians. The prevalence of obesity and related diseases—hypertension, high cholesterol or prediabetes, and smoking—was alarming [12]. The authors emphasized the irrational nutrition model and the urgent need for nutritional education and prevention of civilization-related metabolic diseases among uniformed services. As a consequence of these studies, the National Health Program was conducted in 2016–2020, which included implementing overweight and obesity reduction programs and a comprehensive study of the diet of uniformed services employees.

In light of the above studies, assessing the components of nutritional status seems to be essential both from a health and national security perspective. The aim of the research study was to assess the frequency of abnormalities in the body composition and bone mineralization among officers of uniformed services. Based on previous research, we assumed that we would identify the irregularities mentioned earlier in a significant group of respondents, and as a consequence, individual conversations and training on the importance of leading a healthy lifestyle would be conducted. In this paper, we present the results of the studies realized as a part of the National Health Program concerning candidates and police officers studying and serving in police schools and police training centers.

## 2. Materials and Methods

### 2.1. Participants

The study on elements of nutritional status covered 289 candidates, including 48 female officers and 241 police officers. The study was designed to reflect the characteristics of the entire Police formation as well as the given training unit. Hence, the police officers trained in the Police Academy in Szczytno, Police Training Center in Legionowo, and its subordinate units (the Department of Police Cynology in Sułkowice, Water Police Training Base in Kal, and Police Prevention Department in Rzeszów) were chosen as participants of this study. The study on bone calcification covered 276 police officers trained in Police Training Center in Legionowo, Police Academy in Szczytno, and Police School in Słupsk. The field research was carried out from March to October 2019. The accepted criteria for inclusion in the study included age 18 to 60, active service in Police, and consent to the study. Women declaring pregnancy were excluded from the study.

The following parameters constituting the elements of nutritional status assessment were measured: anthropometric measurements, BMI, and the total fat content in the body. Moreover, bone density was assessed using the DEXA (Dual Energy X-ray Absorptiometry) densitometric method. In the first step, the body weight and height of each of the surveyed officers were measured and then, based on the results, the Body Mass Index (BMI) was determined. The BMI value allowed the subjects to be classified into one of the four groups (according to the WHO classification): underweight (BMI up to 18.4 kg/m^2^), average body weight (BMI 18.5–24.9 kg/m^2^), overweight (BMI 25.0–29.9 kg/m^2^), and obesity (BMI over 30.0 kg/m^2^) [13,14]. Moreover, based on the body fat content measured by multi-frequency bioelectrical impedance, the Fat Mass Index was calculated with the following equation: FMI = fat − free mass/height^2^ (kg/m^2^). The scale of FMI classification developed by Kelly et al. was accepted, adopting FMI values between 3 and 6 as a normal fat mass, FMI < 3 as fat deficit, and FMI > 6 as excess fat [15]. Bone calcification was assessed based on the value of the T-score, in which the mean value and standard deviation in the groups of young adults were adopted as the reference range regardless of the patient’s age. The T-score value −1 was accepted as the normal standard, meaning that it is not smaller than one standard deviation below the mean value. T-score values between −1 and −2.5 are typical for osteopenia, and values less than −2.5 are typical for osteoporosis [16]. The research was conducted in accordance with the 93rd Helsinki Declaration of the World Medical Society and was positively verified by the Ethics Committee of the Military Institute of Hygiene and Epidemiology (No. 1/XXI 95/2016). Participants received an information sheet on the details and the purpose of the study, the procedures used, and the potential risks and benefits of their participation.

### 2.2. Anthropometric Measurements

All measurements were made in accordance with the principles of good practice and procedures specified in the instruction manual by qualified researchers. Height was measured using a portable stadiometer (without shoes) (TANITA HR-001, Tanita Corporation, Tokyo, Japan). The police officers were asked to stand barefoot on the footprints, with his/her heels together and touching the backstop, keeping their legs straight, shoulders relaxed, and head in the horizontal Frankfurt plane position.

Bodyweight and fat content were measured using bioelectrical impedance analysis (BIA) using the TANITA MC-780 103 machine (Tanita Corporation, Tokyo, Japan) with an accuracy of 0.1 kg according to the procedure specified in the instruction manual (lightly dressed, without shoes) and with the use of a standard mode. The measurements were performed at room temperature, in light clothing, and under constant hydration conditions. Subjects were recommended to refrain from eating and intense physical exertion for about three hours before the study. The subjects were asked to discard any transmitting devices such as mobile phones or smartwatches that may affect the readings.

Bone mineral density was measured on the forearm of the non-dominant hand using the DEXA method with the EXA 3000 densitometer (OsteoSys Co., Ltd., Seoul, Korea).

### 2.3. Statistical Analysis

The obtained results are presented as arithmetic means and standard deviation (SD). There were no missing data in the presented study, as the complete set of the presented data was collected from all subjects.

Statistical analyses were performed using Statistica 12.5 software (StatSoft, Tulsa, OK, USA). The Kolmogorov–Smirnov test was used to check the normality of the variables’ distribution. To compare the results between the two groups, the Student’s *t*-test or, in the case of variables with a non-normal distribution, the Mann–Whitney U test was used. For variables with non-normal distribution, non-parametric tests were performed: Kruskal–Wallis test, which is a non-parametric equivalent of one-way analysis of variance, and Dunn’s test, as an equivalent of post hoc tests. The value α = 0.05 was adopted as the level of significance.

## 3. Results

The nutritional status assessment covered 241 officers, including 75 trained at the Police Academy, 69 at the Police Training Center, 39 at the Department of Police Cynology, and 97 at the Police Prevention Unit. The basic characteristics of all surveyed police officers are provided in Table 1, while the detailed breakdown by individual centers is presented in Table 2.

The percentage of normal body weight, overweight and obese police officers in each examined group based on BMI is presented in Table 3.

The basic characteristic of the examined police officers presented in Table 1 showed that examined women were significantly older than men. Moreover, the group of men was characterized by significantly higher body weight and body height, as well as by significantly higher BMI values. Examined female officers were characterized by a higher percentage of body fat.

Statistical analysis performed within the group of men and women (Table 2) revealed significant differences in body height of both men and women depending on the service unit. Trainees in the Department of Police Cynology, both women and men, were significantly shorter in height in relation to their colleagues in the Police Prevention Department and Water Police Training Base, respectively.

The analysis of the body weight and height measurements enabled the calculation of BMI and, in consequence, permitted assigning respondents to appropriate groups according to the WHO classification described in Section 2. None of the surveyed officers were underweight, while over 65% of the examined subjects had increased body weight. These abnormalities were found in one-third of the surveyed women, whereas this percentage was 71.4% in men. According to WHO criteria, 50.2% of all respondents were overweight, including 22.9% women and 55.6% men. Obesity was found in 14.9% of all surveyed officers, including 10.4% of women and 15.8% of men. Such a significant number of obese people in service is alarming since it may contribute to developing cardiovascular diseases and other metabolic civilizational diseases, resulting in their release from service.

BMI does not always provide reliable results for an assessment of body composition, especially in the case of physically active people, since its value may result from high muscle mass and not from high-fat content in the body. Therefore, finding abnormalities in body weight should be based on also on total body fat content. For this reason, we conducted a body adipose tissue measurement to calculate the FMI index. The Fat Mass Index (FMI) calculation ensures a more reliable approach as it differentiates muscle and fat mass of the total body weight. Hence, from a clinical point of view, while assessing anthropometric components of nutritional status, both BMI and FMI should be used as complementary tools. The results of the assessment of the nutritional status of police officers based on the fat content are summarized in Table 4.

Total body fat content (FMI) analysis revealed the presence of excessive fat tissue in 31.8% of all subjects. Overweight and obesity resulting from high-fat content were found in 34.9% of men and 16.7% of women. Consequently, we compared the number of candidates with excess body weight according to the BMI with the measured body fat level. This comparison showed that excessive body mass in 36.5% of men and 16.7% of women results from muscle mass and is not a type of overweight that may be hazardous to health. On the other hand, all subjects with excessive body fat were characterized by abnormalities in BMI.

An adequate supply of calcium is significant for maintaining the proper structure and functioning of the human skeletal system. The content of this mineral, in addition to genetic determinants and the degree of physical activity, is an essential factor influencing bone mass. Therefore, achieving the maximum values of peak bone mass, the main prognostic factor for the risk of osteopenia or osteoporosis is essential in preventing the mentioned bone pathophysiological conditions [17].

The proper status of Police officers’ bone calcification is vital due to the physical burden of the training process and the realization of specific tasks related to this training process. Maximum skeletal calcification, known as peak bone mass, occurs between 25 and 35. This age range is also the period in which most new candidates for service in the Police are admitted. Our study of Police officers’ mineral nutritional status covered 276 officers, including 67 women and 209 men. Bone calcification was assessed based on the T-score value—above or below the expected value for the population of young, healthy people, according to the ranges presented in Section 2.

Densitometric tests revealed that 26.6% of the respondents suffered from disturbances in bone mineralization of varying severity, and when separated by gender, it concerned 52.3% of women and 18.7% of men (Table 5 and Table 6).

The densitometric tests revealed the alteration of bone calcification in female officers regardless of the place of service. We found that 4.6% of the surveyed police officers had changes characteristic of osteoporosis, while bone changes characteristic of osteopenia occurred in 47.7% of the surveyed females. The most significant number of female officers (20) with altered skeletal calcification characteristic of osteopenia was found among women trained at the Police Academy, while the highest percentage (66.6%) was found among women in the Police Training Center (Table 5).

The analysis of the results of densitometric tests in male officers revealed changes in skeletal calcification in all three research sites, and it mainly concerned disorders characteristic of osteopenia (Table 6). The highest percentage of officers with bone density disorders was observed in the Police Academy and the lowest was observed in the Police School in Słupsk.

## 4. Discussion

Since service in the Police is considered a dangerous occupation, it is required that the potential candidate is in excellent health condition [18]. Police officers are a professional group particularly exposed to work-related stress, which may have a negative effect on the health condition. Numerous studies suggest that exposure to both chronic and work-related stress is associated with an increased risk of civilization diseases, including obesity, cardiovascular disease (CVD), and type 2 diabetes (T2DM) [19,20]. Research in this area shows a higher incidence of diseases, especially cardiovascular diseases, among police officers than in other professional groups [21]. This observation is supported by epidemiological reports showing a higher incidence of obesity among police officers in relation to non-police employees [22,23]. Hence, it seems extremely important to study the relationship between body composition and the risk of diet-related metabolic diseases as they may be essential in monitoring the health of police officers. Consequently, obtained observations and results enable the implementation of appropriate preventive measures, which is the main tenet of the National Health Program, partly presented in this paper.

The presented research studies were conducted at the places of service of police officers, which is why we based on measurements performed with mobile equipment. Body composition assessment is a valuable and accessible tool enabling the simple evaluation of physical fitness and health as it results from various factors such as diet, stress, the intensity of physical activity and other daily habits [24,25].

The assessment of nutritional status elements based on anthropometric measurements and BMI revealed excess body weight in the majority of examined Police officers. These results were different depending on the place of the research and ranged from 60.0% to 69.1%. The fact that 71.8% of all male participants and 35.4% of women are overweight is disconcerting. According to the classification of nutritional status based on BMI, 25% of the women and 56% of men were overweight, while obesity was found in 10.4% and 15.8%, respectively. These results align with observations made among uniformed services in recent years, similarly to the civilian population [26].

As we indicated earlier, the assessment of overweight and obesity based on BMI values may not be reliable for adults with increased physical activity and high muscle mass. BMI cannot distinguish between fat mass and lean mass; thus, it is advisable to determine the fat mass and calculate Fat Mass Index (FMI). This indicator assesses the degree of fatness and not the entire body weight, which in the case of a police officer may result in extensive muscle mass, increasing the BMI. The synthesis of those parameters seems to be an optimal approach as it can provide information about the actual number of obese people. The total body fat content measurement in police officers showed that 31.8% of all respondents had increased adipose tissue content. What is essential is that every subject with excessive body fat also had excessive BMI. By sex, irregularities in the body fat content of varying severity were found in 16.7% of female officers and 34.9% of female officers. In our study, the average BMI among Police male officers was above reference values, while the average % fat level and FMI were adequate in both sexes. Our previous research regarding polish soldiers confirmed similar observations [26].

The prevalence of overweight and obesity among police officers is also a significant health problem in other countries. Although the physical demands of police work suggest the importance of maintaining a healthy body weight, studies have shown that 40.5% of U.S. police officers are obese [27]. A similar result was observed in another study of 276 U.S. police officers, where the obesity rate of male police officers was 41.9% [28]. These observations are supported by a study of the nutritional status of 160 Saudi Arabian police officers, where overweight or obesity was found in 66.9% of respondents [29].

Another measure of nutritional status essential for police officers is bone mineral density (calcification). A low value of this parameter leads to osteopenia and osteoporosis. Osteoporosis is a condition that affects the bones, causing them to become weak and fragile and more likely to break (fracture). Changes in bone mass occur due to complex regulatory mechanisms based on genetic factors and the nervous and humoral systems [30]. The role of an adequate supply of calcium and its content in the body for the proper functioning of the skeletal system is indisputable. However, it should be noted that the content of this element in the body undergoes dynamic changes [31]. Therefore, it is essential to achieve the maximum values of peak bone mass, which is the main prognostic factor for the risk of osteopenia or osteoporosis and to monitor bone calcification in the prevention of osteoporosis. This is of particular importance among police officers, as it allows for the early detection of disorders and appropriate therapy and avoiding fractures, which may be the reason for dismissal from service. In the presented study, only half of the examined women revealed a proper BMD (T-score above −1); changes typical of osteopenia were found in 47.7% of subjects, while osteoporosis was found among three of 67 examined women. Significantly better results were obtained in the case of men, as bone calcification disorders concerned only about 20% of the studied population and features of osteoporosis were found in less than 1% of the respondents.

Due to the lack of similar research among police officers, the obtained results can be applied to soldiers who are also subjected to similar conditions of service and training. Our previous research on soldiers trained similarly to candidates in police schools showed that osteopenia and osteoporosis were found in 19.2% of respondents under 30 and 7.4% of soldiers over 30 [32]. On the other hand, studies performed among Polish Land Forces soldiers revealed adequate bone calcification in 75% of soldiers aged up to 30 and among 90% of soldiers aged over 30 [33]. Similarly, 77% of soldiers returning from a mission in Afghanistan, aged up to 30, and 87% of soldiers aged over 30 were characterized by an adequate bone calcification [34].

## 5. Strengths and Limitations

The presented research has several strengths. First, it may be considered a unique study as there are limited numbers of studies among police officers in Poland that assess the elements of the nutritional status (anthropometry, body composition, and bone calcification). Moreover, we focused on three nutritional status indicators: indicators of protein-energy status describing body size (BMI) and obesity (FMI) and indicators of mineral nutritional status (BMD T-score) that are considered essential in the evaluation of police officer’s health and suitability for the service. The results of bone calcification are unique since there are no studies evaluating this aspect of nutritional status among police officers in Poland. Third, based on the body composition and densitometric analysis, we revealed significant abnormalities in Police officers’ nutritional status, which clearly show the necessity to perform educational activities in the health promotion of uniformed services (Police officers). These activities should be primarily focused on nutritional prevention of nutrition-related non-communicable diseases and motivate police officers to respect the basic principles of a healthy lifestyle.

One of the study’s limitations is the relatively small group (289 police officers). However, our research aimed to gain cross-sectional knowledge concerning the tested parameters of officers at their service places. Although the surveyed officers come from different units, this group is a representative sample of this uniformed formation. From the above limitation comes another one concerning using a DEXA-based apparatus for measuring body composition. We agree that this method of collecting data concerning body composition is optimal, but because the research study was carried out off-site in various places of police service throughout the country, we used devices characterized by small size and mobility. We agree that another limitation of the study is the lack of information on the other components of the nutritional status (nutrition and physical activity). However, since the presented work is part of the National Health Program project, these results will be presented in another work comparing research on diet and physical activity in different uniformed services.

## 6. Conclusions

The presented results on elements of the nutritional status of police officers showed the disturbing occurrence of nutritional disorders, especially obesity. This is particularly worrying as it proves an unbalanced nutrition model in terms of energy, predisposing to the development of diet-dependent metabolic civilizational diseases. Moreover, the observed bone calcification disorders in police officers indicate an improper state of mineral nutrition, predisposing them to an increased bone fractures.

The reported abnormalities and disturbances in police officers’ nutritional status suggest the need to train staff responsible for planning and implementing mass catering to optimize the nutrition and nutritional prophylaxis of metabolic diseases. Moreover, the obtained results justify the need for performing educational activities in health promotion and developing a guide on the principles of rational nutrition and prevention of diet-related diseases, taking into account the specific nature of the service in the Police.

## Figures and Tables

**Table 1 ijerph-19-07161-t001:** Basic characteristics of police officers.

	Male	Female	*p*
No. of participants	241	48	
Age (years)	32.6 ± 8.3 ^#^	35.3 ± 9.7 ^#^	0.0471
Body height (cm)	180.2 ± 6.4 ^a^	165.6 ± 5.7 ^a^	0.0000
Body weight (kg)	87.8 ± 13.4 ^a^	67.3 ± 12.0 ^a^	0.0000
BMI Body Mass Index (kg/m^2^)	27.0 ± 3.4 ^a^	24.6 ± 4.7 ^a^	0.0000
Body fat (%)	20.0 ± 6.4 ^a^	27.2 ± 7.6 ^a^	0.0000
FMI Fat Mass Index (kg/m^2^)	5.56 ± 2.36 ^a^	7.00 ± 3.52 ^a^	0.0048

Data are presented as mean ± standard deviation. *p* < 0.05 was adopted as the critical probability value. The results marked with the same symbol (^#^) in the row showed statistically significant differences in Student’s *t*-test. Variables sharing same letter (^a^) are statistically different in Mann–Whitney U test.

**Table 2 ijerph-19-07161-t002:** Characteristics of police officers with the detailed breakdown by individual centers.

Parameter	Sex	Police Academy	Police Training Center	Department of Police Cynology	Water Police Training Base	Police Prevention Department	*p*
Body height (cm)	F	165.8 ± 6.0	166.2 ± 5.9	161.7 ± 3.9 ^a^	-	168.6 ± 7.2 ^a^	0.0412
	M	180.6 ± 6.3	180.5 ± 6.2	178.1 ± 6.57 ^a^	184.5 ± 4.76 ^a^	179.8 ± 6.0	0.0451
Body weight (kg)	F	64.9 ± 10.9	67.3 ± 12.8	71.6 ± 13.4	-	67.3 ± 12.0	n.s.
	M	83.3 ± 10.6	86.5 ± 12.2	88.9 ± 10.6	81.4 ± 9.4	92.1 ± 14.4	n.s.
Body Mass Index(kg/m^2^)	F	23.6 ± 4.0	24.8 ± 5.0	26.6 ± 5.2	-	24.6 ± 4.7	n.s.
	M	25.6 ± 2.5	26.6 ± 3.0	27.4 ± 3.6	25.0 ± 2.1	28.3 ± 3.7	n.s.
Fat Mass Index(kg/m^2^)	F	7.01 ± 3.55	7.10 ± 3.80	8.51 ± 4.02	-	7.01 ± 3.48	n.s.
	M	4.64 ± 1.44	5.27 ± 1.87	5.87 ± 2.14	4.49 ± 1,10	6.55 ± 2.54	n.s.

Data are presented as mean ± standard deviation. *p* < 0.05 was adopted as the critical probability value. The results marked with the same letter in the row showed statistically significant differences in Dunn’s test. M—male; F—female; n.s.—not significant.

**Table 3 ijerph-19-07161-t003:** The number of police officers with normal body weight, overweight, and obesity.

Nutritional Status Parameter	Sex	Total *n* (%)	Police Academy *n* (%)	Police Training Center *n* (%)	Department of Police Cynology *n* (%)	Water Police Training Base Kal *n* (%)	Police Prevention Department Rzeszów *n* (%)
**No. of patricipants**	F	48	15	11	11	0	11
	M	241	70	47	27	11	86
**Normal weight**	F	32	10	9	6	0	7
	M	69	24	10	8	4	23
**Overweight** ***n* (%)**	F	11 (22.9)	3 (20.0)	1 (9.1)	4 (36.4)	0 (0)	3 (27.3)
	M	134 (55.6)	40 (57.1)	24 (51.1)	12 (44.4)	7 (63.6)	51 (59.3)
	Total	145 (50.2)	43 (50.6)	25 (43.1)	16 (42.1)	7 (63.6)	54 (55.7)
**Obesity** ***n* (%)**	F	5 (10.4)	2 (13.3)	1 (9.1)	1 (9.1)	0 (0)	1 (9.1)
	M	38 (15.8)	6 (8.6)	13 (27.7)	7 (25.9)	0 (0)	12 (14.0)
	Total	43 (14.9)	8 (9.4)	14 (24.1)	8 (21.1)	0 (0)	13 (13.4)
**Total** **overweight** **(%)**							
	F	33.3	33.3	18.2	45.5	0	36.4
	M	71.4	65.7	78.7	70.4	63.6	73.3
	Total	65.1	60.0	67.2	63.2	63.6	69.1

M—male; F—female; *n*—number of participants.

**Table 4 ijerph-19-07161-t004:** Assessment of the nutritional status of police officers based on the fat content.

Nutritional Status Parameter	Total *n* (%)			Police Academy Szczytno *n* (%)			Police Training Center Legionowo *n* (%)			Department of Police Cynology Sułkowice *n* (%)			Water Police Training Base Kal *n* (%)	Police Prevention Department Rzeszów *n* (%)		
No. of participants	Σ	F	M	Σ	F	M	Σ	F	M	Σ	F	M	M	Σ	F	M
	289	48	241	85	15	70	58	11	47	38	11	27	11	97	11	86
Normal body fat	198 (68.5)	40 (83.3)	158 (65.6)	65 (76.5)	13 (86.7)	52 (74.3)	37 (63.8)	10 (90.9)	27 (57.4)	24 (63.2)	8 (72.7)	16 (59.3)	9 (81.8)	63 (64.9)	9 (81.8)	54 (62.8)
Excessive body fat	92 (31.8)	8 (16.7)	84 (34.9)	20 (23.5)	2 (13.3)	18 (25.7)	21 (36.2)	1 (9.1)	20 (42.6)	15 (39.5)	3 (27.3)	12 (44.4)	2 (18.2)	34 (35.1)	2 (18.2)	32 (37.2)
Excessive BMI and normal body fat	96 (33.2)	8 (16.7)	88 (36.5)	31 (36.5)	3 (20.0)	28 (40.0)	18 (31.0)	1 (9.1)	17 (36.2)	9 (23.7)	2 (18.2)	7 (25.9)	5 (45.5)	33 (34)	2 (18.2)	31 (36.0)
Excessive BMI and body fat	92 (31.8)	8 (16.7)	84 (34.9)	20 (23.5)	2 (13.3)	18 (25.7)	21 (36.2)	1 (9.1)	20 (42.6)	15 (39.5)	3 (27.3)	12 (44.4)	2 (18.2)	34 (35.1)	2 (18.2)	32 (37.2)

Abbreviations: *n*, number of participants; BMI, body mass index; F, female; M, male; Σ, the sum of the participants of both sexes.

**Table 5 ijerph-19-07161-t005:** Assessment of the bone calcification of female police officers (*n* = 67).

T–Score (Bone Calcification)		Total (*n* = 67)		Police Academy Szczytno (*n* = 40)		Police Training Center Legionowo (*n* = 9)		Police School Słupsk (*n* = 18)	
		*n*	%	*n*	%	*n*	%	*n*	%
<−2.5	Osteoporosis	3	4.6	3	7.5	0	0	0	0
−2.5 to −1	Osteopenia	32	47.7	20	50.0	6	66.6	6	33.3
>−1	Normal	32	47.7	17	42.5	3	33.3	12	66.7

Abbreviations: *n*, number of participants.

**Table 6 ijerph-19-07161-t006:** Assessment of the bone calcification of male police officers (*n* = 209).

T–Score (Bone Calcification)		Total (*n* = 209)		Police Academy Szczytno (*n* = 82)		Police Training Center Legionowo (*n* = 56)		Police School Słupsk (*n* = 71)	
		*n*	%	*n*	%	*n*	%	*n*	%
<−2.5	Osteoporosis	2	1.0	1	1.2	1	1.8	0	0
−2.5 to −1	Osteopenia	37	17.7	20	24.4	12	21.4	5	7.0
>−1	Normal	170	81.3	61	74.4	43	76.8	66	93.0

Abbreviations: *n*, number of participants.

## Data Availability

The data presented in this study are available upon request from the corresponding author. The data are not publicly available due to privacy/ethical restrictions.

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
