# Peer review of "Assessment of the Body Composition and Bone Calcification of Students of Police Schools and Police Training Centers in Poland—A Cross-Sectional Study"

_ijerph, 2022, doi:10.3390/ijerph19127161_

Round 1
Reviewer 1 Report
Substantial revision of the manuscripts is needed.
There are important research outcomes, particularly the idea that the presence of obesity in students of police schools and training centers proves that the present nutrition model is energetically unbalanced, while the demonstrated disorders of bone calcification indicate an improper condition of mineral nutrition. For having such kind of important statement, a comprehensive assessment of nutritional status is needed. Assessing an individual's nutritional status involves not only anthropometrics, but also biochemical, clinical and dietary data. Unfortunately, the study has limitations, which means that substantial revision starting from the title of the manuscripts is needed.
Major comments:
Diet investigation is missing. Study of the diet and nutrient intake is very important in order to assess the balance of both macro and micronutrients.
Introduction must be revised, information related DRV values must be integrated.
The investigation has strong limitations and seems like case study, so it is highly appreciated to involve study limitations and highlight the impact for conclusions.
Author Response
Substantial revision of the manuscripts is needed.
- There are important research outcomes, particularly the idea that the presence of obesity in students of police schools and training centers proves that the present nutrition model is energetically unbalanced, while the demonstrated disorders of bone calcification indicate an improper condition of mineral nutrition. For having such kind of important statement, a comprehensive assessment of nutritional status is needed. Assessing an individual's nutritional status involves not only anthropometrics, but also biochemical, clinical and dietary data. Unfortunately, the study has limitations, which means that substantial revision starting from the title of the manuscripts is needed.
Response: We would like to thank Reviewer 1 for this remark. We have revised the manuscript starting from the title, through the abstract and the content to underline that the presented paper mainly concerns anthropometric measurements, body composition analysis, and the study of bone calcification, which are only elements of the nutritional status assessment.
- Diet investigation is missing. Study of the diet and nutrient intake is very important in order to assess the balance of both macro and micronutrients.
Response: We would like to thank Reviewer 1 for this remark. As suggested by the Reviewer, we emphasised the lack of nutrition assessment in section 5 - Strengths and Limitations of the study.
- Introduction must be revised, information related DRV values must be integrated.
Response: According to the Reviewer’s suggestions, we revised the introduction.
- The investigation has strong limitations and seems like case study, so it is highly appreciated to involve study limitations and highlight the impact for conclusions.
Response: We would like to thank Reviewer 1 for this remark. As suggested, we added a section concerning Strengths and Limitations of the presented study.
Author Response
Reviewer #2:
- (a) Authors should indicate the study’s design with a commonly used term in the title or the abstract: cross sectional.
(b) Provide in the abstract an informative and balanced summary of what was done and what was found. This abstract look attractive. The English language should be edited carefully. Also, why you are nominating the study as : the protein and energy nutritional status assessment study? There is no need to provide a name for such studies. You can mention only survey assessing .. The first sentence in the abstract should be replace by another one showing an introduction of your topic. You need to enrich your abstract in term of style and English. ”Two hundred eighty-nine students of Polish police 19 schools and police training centres were included in the protein and energy nutritional status assessment study, while the mineral nutritional status was assessed in 276 people trained in 4 police 21 schools and training centres” this sentence is unclear. Do you mean you have 2 study groups? And what is : the mineral nutritional status. Please re-write all the abstract in a way that reflect your findings.
Response: We would like to thank Reviewer 2 for these remarks. We have revised the title and the abstract as suggested.
- There is no scientific background in the introduction that reflects the title, abstract
and the inclusion of study participants. Authors were only talking about. Obesity, fatness, overweight and their health impact. The rationale for the investigation being reported is missing. Consequently, the introduction paragraph should be repeated accordingly. It should contain rationale for selection of such type of population and what is the health impact of being overweight or obese or having low bone mineral density among these people.
- Authors should state their specific objectives, including any prespecified hypotheses in lines 75
Response: We would like to thank Reviewer 2 for these remarks. We have revised the introduction section according to Reviewer 2 suggestions.
- (a) It is better to modify this title: The protein-energy nutritional status assessment study in line 78. There are some ambiguities in the data collection: I am wondering why the fatness indication was BMI for such population? I think the authors missed the concept that BMI cannot be used as indicator for health assessment in people who can be muscular. Do the authors know if this population were muscular or have a regular physical activity patterns or athletes? Because BMI doesn’t gauge excess fat in these people. Moreover, if the authors used DEXA to assess the bone density status of their participants, why the analysis of the body composition wasn t based on this machine? DEXA is the gold measurement tool that show adiposity, muscle mass and bone density. This makes all the study flaw.
Response: We would like to thank Reviewer 2 for these remarks. We have revised the methods part with an underlined justification for choosing BMI and FMI to determine the fatness of the population, as well as we added section 5 (strengths and limitations), where we explained the rationale behind choosing the BIA method of body composition analysis.
- The authors should describe the setting, locations, and relevant dates, including periods of recruitment, exposure, follow-up, and data collection
Response: We would like to thank Reviewer 2 for these remarks. Information about the location of the service sites and the period of field research were added.
- All outcomes are well described and the assessment methods are described but all in all, these variables reflect the scientific interpretation of the topic.
- Authors should describe any efforts to address potential sources of bias in their data
- Authors should explain how the study size was arrived at.
- Explain how quantitative variables were handled in the analyses. Authors should describe which groupings were chosen and why. Authors did not show any analysis between genders. Where is the p-values in table 1?
Response: We would like to thank Reviewer 2 for these remarks. We have revised the methods section according to Reviewer 2 suggestions.
- (a) Authors should describe all statistical methods, including those used to control for confounding
(b) Authors should describe any methods used to examine subgroups and interactions
(c) Authors should explain how missing data were addressed
Response: We would like to thank Reviewer 2 for this remark. We have added the description of the used statistical methods. Information about the lack of missing data was also added.
- (a) Authors should report numbers of individuals at each stage of study eg. numbers potentially eligible, examined for eligibility, confirmed eligible, included in the study, completing follow-up, and analysed
(b) Authors should give reasons for non-participation at each stage
(c) Authors should consider use of a flow diagram
Response: We would like to thank Reviewer 2 for these remarks. We have added the information numbers of examined individuals from different service sites. Since the only exclusion criteria was pregnancy, subjects declaring pregnancy were utterly excluded from the study from the very beginning.
- (a) Line 111-112: All measurements were performed according to the procedure specified in the instruction manual and without any metal objects. Please clarify.
(b) Please indicate number of participants with missing data for each variable of interest
Response: We would like to thank Reviewer 2 for these remarks. A detailed description of measurement methodology and information about missing data were added.
- The authors reported the values obtained following the analysis. However, how the authors interpreted these values if they are not comparing between the sub groups. What is the aim behind identifying only the percentage of overweight or obese people? Where are the correlates and causes behind this obesity. There is no odd ratio or any correlational analysis in this study. This makes the whole data un-useful.
Response: We would like to thank Reviewer 2 for these remarks. The assumption of the presented study was to investigate the frequency of overweight, obesity and bone calcification disorders in uniformed services and, on their basis, to implement appropriate educational and preventive programs. For this reason, we have adopted the form of presentation of the obtained results as the percentage frequency of these disorders. However, on the reviewers' advice, we performed basic statistical analyses, the results of which are presented in Tables 1 and 2.
- Line 181-182 : Summarise key results with reference to study objectives
Lines 186 -190: authors are depending on BMI to show possible CVD risk? What is
your population were muscular?
Lines 205-213: these are the limits of the study and this is the reason for the weakness of the whole manuscript. The whole discussion is very weak
- Discuss limitations of the study, taking into account sources of potential bias or imprecision. Discuss both direction and magnitude of any potential bias
Response: We would like to thank Reviewer 2 for these remarks. We have revised the discussion section and added the description of limitations according to Reviewer 2 suggestions.
- Authors should give a cautious overall interpretation of results considering objectives, limitations, multiplicity of analyses, results from similar studies, and other relevant evidence Discuss the generalisability (external validity) of the study results Authors should have one paragraph as conclusion and not listing information as points
Response: We would like to thank Reviewer 2 for these remarks. We have revised the results section according to Reviewer 2 suggestions.
Reviewer 3 Report
The introduction is very general since it considers a global and world problem that is the relationship between overweight and obesity, plus the high percentage of body fat with chronic diseases. Something similar is observed in the relationship between mineral intake and osteoporosis and osteopenia. It does not specify the importance of the study in police centers, it is not justified correctly, it does not declare the importance of the investigation.
In method, the type of study is not described, nor is the determination of the sample, statistical analyzes are not mentioned, and hypotheses are not considered. Fundamental procedures of the measurements carried out are omitted. It is not enough to follow the instruction manual, qualified and trained personnel are required to take measurements.
The results are very basic, comparisons are described and there is no inferential statistical test which could improve the focus of the study.
In discussion, comparisons are made with few references and it is not clear finally the importance of determining BMI and BODY FAT being a population with high physical activity and therefore high muscle mass. It is striking that food intake and physical activity are considered here, but they were never measured or at least declared.
The conclusions are very basic, nor does it respond to a hypothesis.
Study requires new analysis and major changes in Introduction, methods, results and conclusions.
Author Response
Reviewer #3:
- The introduction is very general since it considers a global and world problem that is the relationship between overweight and obesity, plus the high percentage of body fat with chronic diseases. Something similar is observed in the relationship between mineral intake and osteoporosis and osteopenia. It does not specify the importance of the study in police centers, it is not justified correctly, it does not declare the importance of the investigation.
Response: We would like to thank Reviewer 3 for this remark. We have revised the manuscript starting from the title to the abstract and the content, including the introduction. We underlined the importance and justification for presented studies.
- In method, the type of study is not described, nor is the determination of the sample, statistical analyses are not mentioned, and hypotheses are not considered. Fundamental procedures of the measurements carried out are omitted. It is not enough to follow the instruction manual, qualified and trained personnel are required to take measurements.
Response: We would like to thank Reviewer 3 for this remark. We have revised the manuscript starting from the title, through the abstract and the content including introduction as suggested by the Reviewer.
- The results are very basic, comparisons are described and there is no inferential statistical test which could improve the focus of the study.
Response: We would like to thank Reviewer 3 for this remark. We have revised the results section and added a statistical test as suggested by the Reviewer.
- In discussion, comparisons are made with few references and it is not clear finally the importance of determining BMI and BODY FAT being a population with high physical activity and therefore high muscle mass. It is striking that food intake and physical activity are considered here, but they were never measured or at least declared.
Response: We would like to thank Reviewer 3 for these remarks. We have revised the discussion section by adding justification for the use of BMI and body fat (FMI). Also, we added a section concerning Strengths and Limitations of the presented study with the explanation addressing Reviewer 3 remark concerning nutrition and physical activity.
- The conclusions are very basic, nor does it respond to a hypothesis.
Response: We would like to thank Reviewer 3 for this remarks. The conclusion section was modified in accordance with the Reviewer’s suggestion.
- Study requires new analysis and major changes in introduction, methods, results and conclusions.
Response: We would like to thank Reviewer 3 for these remarks. We have revised every section of the manuscript, and we do hope Reviewer 3 will find them suitable and sufficient.
Round 2
Reviewer 1 Report
The authors made a significant changes and the paper can be accepted
Reviewer 2 Report
The authors replied to all comments and the manuscript shaped well
Reviewer 3 Report
The article is acceptable, suggested corrections were made.Execute what was marked by another evaluator